# The Evaluation of Functional Abilities Using the Modified Fullerton Functional Fitness Test Is a Valuable Accessory in Diagnosing Men with Heart Failure

**DOI:** 10.3390/ijerph19159210

**Published:** 2022-07-28

**Authors:** Magdalena Migaj, Marta Kałużna-Oleksy, Jacek Migaj, Anna Straburzyńska-Lupa

**Affiliations:** 1Department of Physical Therapy and Sports Recovery, Poznan University of Physical Education, 61-871 Poznań, Poland; straburzynskalupa@awf.poznan.pl; 21st Department of Cardiology, Heliodor Swiecicki Clinical Hospital, Poznan University of Medical Sciences, 61-848 Poznań, Poland; marta.kaluzna-oleksy@skpp.edu.pl (M.K.-O.); jacek.migaj@skpp.edu.pl (J.M.)

**Keywords:** handgrip strength, heart failure prognosis, 6-min walk test, cardiopulmonary exercise test

## Abstract

The assessment of functional abilities reflects the ability to perform everyday life activities that require specific endurance and physical fitness. The Fullerton functional fitness test (FFFT) seems to be the most appropriate for assessing physical fitness in heart failure (HF) patients. The study group consisted of 30 consecutive patients hospitalized for the routine assessment of HF with a reduced ejection fraction (HFrEF). They formed the study group, and 24 healthy subjects formed the control group. Each patient underwent a cardiopulmonary exercise test (CPET), transthoracic echocardiography and FFFT modified by adding the measurement of the handgrip force of the dominant limb with the digital dynamometer. The HF patients had significantly lower peak oxygen uptake (peakVO_2_), maximal minute ventilation, and higher ventilatory equivalent (VE/VCO_2_). The concentrations of B-type natriuretic peptide (BNP) and N-terminal proBNP (NT-proBNP) were significantly higher in the study group. The results of all the FFFT items were significantly worse in the study group. FFFT parameters, together with the assessment of the strength of the handgrip, strongly correlated with the results of standard tests in HF. FFFT is an effective and safe tool for the functional evaluation of patients with HFrEF. Simple muscle strength measurement with a hand-held dynamometer can become a convenient and practical indicator of muscle strength in HF patients.

## 1. Introduction

Heart failure (HF) is a growing clinical, economic, and epidemiological problem. Despite the progress in the treatment, the prognosis is still poor. It is estimated that during the prevailing SARS-CoV-2 pandemic the number of new HF cases may significantly increase both in Europe and worldwide [1].

Heart failure is a progressive and systemic disease characterized by a reduced ability to perform functional activities. Climbing stairs, walking, moving around, cleaning, and bathing are the most burdensome problems in heart failure patients’ everyday activities [2]. Patients also have problems bending down [3] and lifting heavy objects [4]. The assessment of functional abilities reflects the ability to perform everyday life activities that require specific endurance and physical fitness [5].

The gold standard for assessing heart failure patients is the cardiopulmonary exercise test (CPET) [6]. This is an exercise test with assessment of the parameters of ventilation and gas exchange, which allows for a detailed analysis of the capacity of the cardiovascular, respiratory and motor systems. CPET is primarily used in the diagnostic workup of reduced exercise tolerance, qualification for heart transplantation and in training planning and evaluation in professional athletes. A 6-min walk test (6MWT) plays a similar role in severe heart failure patients who are not able to complete a regular exercise test. Reduced exercise capacity, a consequence of progressive heart failure, causes a gradual deterioration of the patient’s condition, limiting their physical activity and, consequently, physical fitness [7]. The Fullerton functional fitness test (FFFT) reflects exercise capacity and fitness [5]. At the same time, CPET and 6MWT, despite their proven value, do not fully assess the problems associated with the daily activities of heart failure patients.

The natriuretic peptides are used widely in the diagnostic workup of heart failure [6]. They indicate overload of the left ventricle, and high values of B-type natriuretic peptide (BNP) and N-terminal B-type natriuretic pro-peptide (NT-pro BNP) may be associated with the severity of heart failure.

The Fullerton functional fitness test seems to be the most appropriate for assessing physical fitness in heart failure patients. Initially, this test was designed to assess the physical fitness of the elderly, i.e., those aged 60–94 [8]. Over time the test has been started to evaluate the results of rehabilitation in various diseases such as chronic obstructive pulmonary disease (COPD) [9], obesity [10], diabetes [11], Parkinson’s disease [12], Alzheimer’s disease [13] and many others [14,15,16]. The Fullerton functional fitness test assesses the key elements of physical fitness: muscle endurance, aerobic capacity, flexibility, agility, and balance. At the same time, this test does not expose the patient to the risk of injuries [17]. Furthermore, the repeatability of each test and the simplicity of its execution enable the test to be carried out by a physiotherapist in hospital and outpatient settings [7].

We aimed to evaluate the usefulness of FFFT in the functional assessment of physical fitness of heart failure patients. In particular, we aimed to correlate FFFT results with CPET and echocardiography results as well as natriuretic peptide values.

## 2. Materials and Methods

### 2.1. Study Population

Fifty-four men were included in the study. The study group consisted of 30 consecutive patients hospitalized for the routine assessment of heart failure with a reduced ejection fraction. The control group consisted of 24 men with no history of heart failure and normal left ventricular ejection fraction (LVEF). The inclusion and exclusion criteria are shown in Table 1.

Each patient underwent CPET and transthoracic echocardiography. NYHA functional class, hemoglobin concentration, B-type natriuretic peptide (BNP) and N-terminal B-type natriuretic pro-peptide (NT-pro BNP) levels were also assessed.

This research was approved by the Bioethics Committee at the Medical University of Karol Marcinkowski in Poznan, number 630/15.

### 2.2. Fullerton Functional Fitness Test

According to commonly accepted procedures, physical fitness was assessed using FFFT (Table 2) [18]. For the safety of patients, the test in both groups was performed by a physiotherapist in the presence of a cardiologist. Before and after the start of the study, the values of heart rate and blood pressure were measured. To complete the assessment of patients with heart failure, the FFFT was modified by adding the measurement of the handgrip force of the dominant limb with the digital dynamometer CHARDER MG-4800, Taiwan.

### 2.3. Statistical Analysis

Statistical analysis was performed using the Statistica 13.1 software. According to the results of the Shapiro-Wilk test, the distribution of all examined variables was considered non-parametric. The Mann-Whitney U test was used to analyze independent continuous data, and the sign test was used to analyze dependent continuous variables. Spearman’s correlation coefficients were calculated to evaluate the relationship between pairs of selected parameters in the heart failure group. *p* < 0.05 was considered significant for all tests.

## 3. Results

The basic characteristics of the study groups are presented in Table 3. The mean left ventricular ejection fraction (LVEF) in the study group was 23.0 ± 6.2%. Moreover, heart failure patients had a significantly larger left ventricular end-diastolic diameter, right ventricular diameter, and left atrium diameter. The mean peak oxygen uptake in the test group was 18.3 ± 5.6 mL/kg/min, and in the control group, this was 33.5 ± 8.1 mL/kg/min. Heart failure patients had significantly lower VCO_2_, maximal minute ventilation, peakCO_2_, and higher ventilatory equivalent VE/VCO_2_. The concentrations of BNP and NT-proBNP were significantly higher in the study group.

Significant differences were observed in the Fullerton functional fitness test results between the study and the control groups (Table 3). Two heart failure patients could not complete FFFT due to rapidly increasing fatigue, but no pathological symptoms of exercise intolerance were observed. All results were significantly worse in the study group. Moreover, two heart failure patients completed the 6-min walk test after 2.36 min and 4.26 min due to very large fatigue and pain in the lower limbs.

Spearman’s correlation analysis (Table 4) was performed to assess the relationship between FFFT including handgrip strength with left ventricular ejection fraction (LVEF), and peak oxygen consumption (peakVO_2_). FFFT parameters, together with the assessment of the strength of the handgrip, strongly correlated with the results of standard tests in heart failure.

## 4. Discussion

Our study is the first analysis of the functional performance assessment of heart failure patients using the Fullerton functional fitness test, in which, the results have been correlated with recognized parameters for assessing poor prognosis in patients with heart failure with reduced ejection fraction, such as LVEF, NT-proBNP, and cardiopulmonary stress test results.

Despite the need to assess the physical fitness of heart failure patients, the available literature is dominated mainly by the assessment of fitness and quality of life [20,21]. Functional abilities are primarily assessed with the help of the 6-min walk test, which does not assess the basic characteristics of physical fitness such as muscle strength, endurance, coordination, and flexibility.

Reduced exercise capacity, which is a consequence of progressive heart failure, causes a gradual deterioration of the patient’s condition, limiting physical activity and, consequently, physical fitness [7]. The Fullerton functional fitness test reflects changes in physical capacity and fitness [5], while a cardiopulmonary stress test and a 6-min walk test, despite their proven value, do not fully assess the problems associated with the daily activities of heart failure patients. Therefore, in our research, the Fullerton functional fitness test was used for the initial assessment of functional fitness, which reflects changes in physical capacity and fitness [5].

Muscular endurance is the ability to perform everyday activities without enormous and rapidly increasing fatigue [22]. Muscle endurance depends largely on the body’s endurance, which in people with heart failure is up to about 50% lower than appropriate [23]. In our research, the patients with heart failure obtained significantly worse results in the endurance-related FFFT items than the controls. There were positive correlations of both the “Chair Stand” and “Arm Curl” tests with LVEF and peakVO_2_ and a negative correlation with NT-proBNP. This can be interpreted as the decreased muscle endurance of lower and upper limbs with progressing heart failure. Our results are consistent with the results of a similar analysis by Węgrzynowska-Teodorczyk et al. [24].

The heart failure hallmark is exercise intolerance, manifested by shortness of breath and a feeling of quickly becoming fatigued. When the disease is highly advanced, symptoms occur even with minimal physical exertion or at rest [6]. Cardiopulmonary exercise test (CPET) on a treadmill or cyclo-ergometer is the “gold standard” for assessing exercise capacity, but it can be difficult for the patient to perform, and its availability is not common.

A healthy person will not achieve maximum effort during a 6-min walk test, but in severely impaired people who cannot perform significant effort during a cardiopulmonary exercise test, a 6-min walk test may be the maximal test. In our study, heart failure patients obtained significantly worse results in both tests. Worse results of a 6-min walk test are associated with an unfavorable prognosis [25].

Our observations are consistent with previously published studies in which patients with severe heart failure (NYHA III-IV) had a much shorter 6-min walk test distance than the controls (NYHA II) [26]. As expected, significant positive correlations between the 6-min walk test distance, LVEF, and peak oxygen consumption (peakVO_2_) were observed in heart failure patients. Previously, a linear correlation between the 6-min walk test distance and the measurement of peak oxygen consumption was demonstrated and had a significant prognostic value in heart failure [8]. The consistency of these findings was expected because of the characteristics of heart failure patients. The progression of heart failure is associated with the reduction of cardiac output (evaluated using LVEF and peakVO_2_) and left ventricular overload (evaluated by BNP and NT-proBNP blood concentrations), which is followed by a reduction in exercise tolerance.

Moreover, like other authors, we observed a significant negative correlation between NT-proBNP levels and the 6-min walk test result [27]. However, other studies also found an increase in physical capacity and a decrease in NT-proBNP levels, which correlated with an improvement in the clinical condition assessed with the NYHA functional class in patients with heart failure participating in physical training [28].

Increased fatigue and muscle weakness in patients with heart failure and the adverse effects of some drugs leading to bradycardia and orthostatic hypotension make patients more prone to falls, especially those over 65 years of age [29,30]. Our study shows that heart failure patients have significantly poorer agility and dynamic balance, contributing to an increased risk of falls. Węgrzynowska-Teodorczyk et al. obtained similar results [31], and worse motor skills of heart failure patients were confirmed using magnetic resonance imaging of the head [32]. The assessment of agility and dynamic balance seems to be important in the initial evaluation of heart failure patients because up to 43% of patients report frequent falls [33].

In our study, the heart failure patients performed worse than controls in the flexibility-related FFFT items. However, a similar assessment of flexibility [24,31] showed contradictory results. Some authors state that flexibility is a morphological, genetically determined trait and is not related to the progression of the disease [34,35]. The difference between the results of the above-mentioned papers and our own may be due to the more significant advancement of the disease determined by LVEF in our study.

Loss of muscle strength and mass may be a consequence of chronic diseases, leading to changes in metabolism and muscle structure [36]. It contributes to the deterioration of the patient’s clinical condition and exercise tolerance [37]. The strength of the knee flexors and extensors in heart failure patients was investigated by Toth et al. [38] who determined that the maximum measurement of the muscle strength of the lower limbs was lower by 15–33% in the group of heart failure patients [38]. Other authors obtained similar results in assessing the muscle strength of the lower extremities [37,39]. It is also assumed that a reduction in the strength of the quadriceps muscle of the thigh may have a prognostic value [37,39] and contribute to a reduction in the quality of life of heart failure patients [40]. The handgrip strength strongly correlates with other measures of muscle strength [37,41]. Our research assessed the strength of the dominant handgrip, which was lower in the group of heart failure patients. Additionally, the analysis of our results showed a significant positive correlation between the handgrip strength and LVEF as well as peakVO_2_, which is consistent with the results of other authors [42]. The observed loss of muscle strength may confirm one of the hypotheses indicating the role of peripheral (muscle) factors in exercise intolerance, which concerns structural, metabolic, or functional changes in skeletal muscles in the course of heart failure [36,37,43]. We chose to examine the dominant limb, similarly to other authors [42].

### Study Limitations

The relatively small size of the study group, and considering only men, may make it difficult to generalize the results to the entire population of heart failure patients. We did not evaluate the possible impact of obesity on the Fullerton fitness test results and on the concentrations of natriuretic peptides. Therefore, it may be necessary to continue the research taking into account obesity indicators (percentage and distribution of adipose tissue in the body).

## 5. Conclusions

The Fullerton fitness test is an effective and safe tool for the functional evaluation of patients with heart failure with reduced ejection fraction. The Fullerton fitness test results correlate well with the results of standardized testing in heart failure patients (most of the FFFT items show positive correlations with left ventricular ejection fraction and negative correlations with BNP and NT-proBNP values). Simple muscle strength measurement with a hand-held dynamometer can become a convenient and practical indicator of muscle strength in heart failure patients.

## Figures and Tables

**Table 1 ijerph-19-09210-t001:** Inclusion and exclusion criteria.

	Inclusion Criteria	Exclusion Criteria
Study group	Age > 18 years Written consent to inclusion in the study Chronic heart failure classified as New York Heart Association (NYHA) functional class II or III reduced left ventricular ejection fraction (LVEF ≤ 40%)	HF exacerbation requiring intravenous diuretics during the 4 weeks before study enrollment Acute coronary syndrome Motor dysfunction preventing completion of the Fullerton fitness test items Uncontrolled blood pressure exceeding 160/100 mmHg at rest
Control group	Age > 18 years Written consent to inclusion in the study	Chronic or acute heart failure Motor dysfunction preventing completion of the Fullerton fitness test items Acute coronary syndrome Uncontrolled blood pressure exceeding 160/100 mmHg at rest

**Table 2 ijerph-19-09210-t002:** Six motor tasks assessed in the Fullerton functional fitness test.

**8 Foot Up&Go**	The patient circles the cone in the shortest possible time at a distance of 2.44 m from the sitting starting position and returns to the starting position.
**30-Second Chair Stand**	The patient repeats full stands from the sitting position. Repetitions are performed within 30 s with the arms crossed over the chest.
**Arm Curl**	The patient flexes the forearm with a 3.5 kg weight in 30 s. The result is the number of repetitions.
**Back Scratch**	The patient tries to join the hands behind the back, leading one hand from the top, and the other from the bottom. The result given in centimeters indicates the distance between the middle fingers. The value may be negative when the patient reaches further than the fingertips.
**Chair Sit&Reach**	From a sitting position on a chair, the patient tries to reach the toes with the leg straight in the knee joint. The result in centimeters shows the distance between the fingers and the toes. The value can be negative when the patient is out of range of motion.
**6-Min Walking Test (6MWT)**	The test result is the number of meters the patient walked along a 30-m corridor in 6 min.
**The modification of the Fullerton functional fitness test:**	
**Measuring the strength of the handgrip**	The examination was performed on the dominant limb, in a sitting position, with the elbow extended and the shoulder joint flexed to 90°. Three measurements were made with a 5-s interval between attempts, and the best measurement was selected for analysis, according to the previously proposed methodology [19].

**Table 3 ijerph-19-09210-t003:** Baseline study population characteristics.

	HF Group X(SD)	Control Group X(SD)	*p*
N Age (years)	30 56.2 (12.2)	24 55.4 (10.4)	0.60
BMI (kg/m^2^) NYHA	28.5 (4.0) 2.4 (0.5)	25.6 (34) -	0.004 -
LVEF (%)	23.0 (6.2)	61.6 (3.6)	<0.001
LVED (mm)	72.4 (8.2)	47.2 (5.9)	<0.001
RVD (mm)	35.4 (7.2)	29.2 (3.5)	<0.001
LAD (mm)	49.8 (10.4)	36.3 (4.3)	<0.001
IVS (mm)	9.6 (1.6)	10.0 (1.0)	0.22
LVPW (mm)	9.9 (1.1)	9.9 (0.9)	0.77
peakVO_2_ (%)	55.8 (12.6)	100.6 (25.5)	<0.001
peakVO_2_ (mL/kg/min)	18.3 (5.6)	33.5 (8.1)	<0.001
peakVO_2_ (L/min)	1.6 (0.5)	2.8 (0.7)	<0.001
peakVCO_2_ (L/min)	1.6 (0.5)	3.1 (0.8)	<0.001
VE Max (L/min)	60.7 (15.7)	91.1 (22.5)	<0.001
peakVCO_2_ (mL/kg/min)	33.7 (6.8)	40.8 (4.5)	<0.001
RER	1.3	1.1	
VE/VCO_2_ slope	32.5 (7.3)	24.0 (3.5)	<0.001
BNP (pg/mL)	384.9 (403)	39.3 (58.1)	<0.001
NT-proBNP (pg/mL)	1823.0 (18301.1)	139.9 (261.5)	<0.001
HGB (mmol/L)	9.0 (0.8)	9.0 (0.8)	0.84
Fullerton functional fitness test results			
6MTW (m)	363.6 (125.1)	563.8 (69.9)	>0.001
Chair Stand (repetitions)	12.7 (5.2)	18.0 (4.5)	>0.001
Arm Curl (repetitions)	14.7 (4.4)	23.3 (4.9)	>0.001
Chair Sit&Reach (cm)	−15.7 (12.1)	−6.8 (8.4)	0.004
Back Scratch (cm)	−20.1 (16.4)	−6.2 (13.6)	0.002
8-foot Up&Go (s)	8.7 (2.2)	6.0 (1.1)	>0.001
Handgrip strength (kg)	37.9 (10.7)	48.3 (10.7)	>0.001

6MWT—6-min walk test; BMI—body mass index; LVEF—left ventricular ejection fraction; HF—heart failure; BNP—natriuretic peptide; HGB—hemoglobin; LVED—left ventricular end-diastolic diameter; RVD—right ventricular diameter; LAD—left atrium diameter; IVS—ventricular septum diameter; LVPW—left ventricular posterior wall diameter; VO_2_—oxygen consumption during exercise; VCO_2_—production of carbon dioxide; VEmax—maximum minute ventilation; peakCO_2_—peak carbon dioxide release; RER—respiratory rate; VE/VCO_2_—exercise ventilation.

**Table 4 ijerph-19-09210-t004:** The correlation between the Fullerton functional fitness test results including handgrip strength with LVEF, peakVO_2_ and NT-proBNP values.

	LVEF		peakVO_2_		NT-proBNP	
	r	*p*	r	*p*	r	*p*
6MTW	0.76	<0.001	0.83	<0.001	−0.71	<0.001
Chair Stand	0.63	<0.001	0.60	<0.001	−0.49	<0.001
Arm Curl	0.64	<0.001	0.76	<0.001	−0.50	<0.001
Chair Sit&Reach	0.38	0.005	0.36	0.008	−0.22	0.72
Back Scratch	0.46	<0.001	0.61	<0.001	−0.36	<0.001
8-foot Up&Go	−0.69	<0.001	−0.77	<0.001	0.57	<0.001
Hand grip strength	0.39	0.004	0.45	<0.001	−0.50	<0.001

6MWT—6-min walk test; LVEF—left ventricular ejection fraction; NT-proBNP—N-terminal brain natriuretic pro-peptide; peakVO_2_—peak oxygen uptake.

## Data Availability

The data presented in this study are available on request from the corresponding author.

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
