# Peer review of "The Evaluation of Functional Abilities Using the Modified Fullerton Functional Fitness Test Is a Valuable Accessory in Diagnosing Men with Heart Failure"

_ijerph, 2022, doi:10.3390/ijerph19159210_

Round 1

Reviewer 1 Report

Thank you for allowing me to review this great work by the authors on "The evaluation of functional abilities using the modified Fullerton functional fitness test is a valuable accessory in diagnosing patients with heart failure". 

This is an interesting topic and the authors did a good job explaining the scientific merits behind doing this study. Despite the small number of subjects in the study as the authors alluded to at the end, it still can provide a value to the readers.

Critique:

- Table 1 and 2 can be merged into one table. Also would suggest using ( ) in the column to the left instead of [ ].

- The discussion is too long and seems very redundant. The authors did not have to discuss each item of the test separately. I think the discussion section needs to be rewritten with more concise narrative that describes their findings and describe the clinical significance while addressing other similar studies and projects.

Thank you.

Author Response

Thank you for all your remarks. We appreciate your helpful review.

1. As you suggested, we merged the indicated tables.

2. You suggested that the discussion needs significant changes. Simultaneously, the other reviewers specifically praised the discussion in its current form. Therefore, we would ask you to confirm that the discussion needs such an extensive revision - we are ready to change it, if you believe it necessary.

Thank you once again

Reviewer 2 Report

Thank you to the authors for submitting their manuscript to International Journal of Environmental Research and Public Health; I enjoyed reading it. 

There some suggestions that I think will provide clarity to the reader (outlined below).

Good luck with your amendments and I look forward to seeing the revised version. 

TITLE

Include “male or men”

SPECIFIC COMMENTS

It is necessary to reduce the high number of acronyms throughout the text to make easier to read it. 

ABSTRACT

Define HF, CPET, VCO2, VE/VCO2. Review all the abstract.

Keywords: 

Define HF and 6MWT 

Introduction:

This section needs to be improved. It is necessary to show more information to the readers. 

Provide more information about this CPET test (line 38)

Add more information about this interesting idea (line 39-40)

Could you provide information about the reliability of this test in different populations? (line 45)

Add these natriuretic peptide values (line 57)

Materials and Methods:

Include the Declaration of Helsinki and the code number of the Ethical Committee in this section too. 

Define NYHA and LVEF. 

Include more information about digital dynamometer like city and country. (line 78)

Why did not you evaluate non-dominant limb with the digital dynamometer? (line 78)

P in lowercase and italic (line 88)

Results

Include a new table with sample characteristics in "study population" section. In this table (table 2) include the rest of variable and change it the name. (line 99)

Discussion: 

The discussion section is clear and well-written. 

Show results (line 160)

Show results (line 200)

Why did they show the same results than yours? Discuss it

Author Response

Thank you for your remarks. We appreciate your helpful comments.

We followed all your suggestions and made the appropriate changes.

Reviewer 3 Report

Migaj_Fullerton test_ijerph_2022

I commend the authors on the completion of this manuscript. 

The article includes a comprehensive introduction and background. This section is sufficient to demonstrate the justification for the development of the study in the field of knowledge.

The research question is well defined, being clinically relevant. The presentation defines the research question. 

But I have some concerns highlighted below.

Abstract

Please, avoid the use of unusual acronyms as CPET, FFFT, BNP and NT-proBNP.

Methods

2.1. Study population. 

Line 63: NYHA. Please explain the acronym the first time it appears in the text: “NYHA functional class II or III, and re-63”.

Line 68: “Each patient underwent CPET”. Please, is it possible to explain a little more about the type of cardiopulmonary exercise test performed? 

Line 70: “N-terminal B-type natriuretic propeptide (NT-pro BNP) levels were assessed.” Please, add hemoglobin that was also assessed. 

Please, include some statement explaining how the consent of the patients was required.

2.3. Statistical analysis

Line 86. “and the sign test was used to analyze dependent continuous variables.” I think you have not used the sign test to analyze dependent continuous variables. I think that all the comparisons have been made in independent continuous data with the Mann-Whitney U test. 

Line 86: “Spearman's 86 correlation coefficients were calculated to evaluate the relationship between pairs of se-87 lected parameters.” Please change to: “Spearman's 86 correlation coefficients were calculated to evaluate the relationship between pairs of se-87 lected parameters in the HF group”. 

Results

Line 143: “Spearman's correlation analysis (Table 4) was performed to assess the relationship 143 between FFFT and handgrip strength, left ventricular ejection fraction (LVEF), and peak 144 oxygen consumption (peakVO2).” Please change to “Spearman's correlation analysis (Table 4) was performed to assess the relationship 143 between FFFT including handgrip strength with left ventricular ejection fraction (LVEF), peak 144 oxygen consumption (peakVO2) and NT-proBNP.” Please syntax review also in the legend of the Table 4.

Table 4. Are you sure that rho spearman sign values are correct for Chair Sit&Reach and Back Scratch? 

Discussion

Line 175: “Patients with HF ob-175 tained significantly worse results in both trials than in the control group. Additionally, 176 positive correlations of both the "Chair Stand" and "Arm Curl" tests with LVEF and 177 peakVO2 and a negative correlation with NT-proBNP were observed. This can be inter-178 preted as the increased muscle endurance of lower and upper limbs with progressing HF. 179” Please extensive syntax review. Muscle endurance is decreased with progressing HF. 

Line 255: “The own research assessed the strength of the dom-255”, please change to “our research assessed the strength…”

Line 274: “No 274 similar dependence was obtained in the author's own work, which may be due to the 275 group's small size.” Please, remove this sentence, you have not analyzed the influence of age or NYHA functional class in this study. 

Line 280: “In the presented 280 study, obesity indicators and their possible impact on the FFFT results and the concentra-281 tion of natriuretic peptides were not considered.” Please, syntax review. 

Author Response

Thank you for all your remarks. We appreciate your helpful comments.

We followed all your suggestions and made the appropriate changes.

To answer your questions regarding the statistics:

  1. the description of the statistical tests was prepared for us by our statistician, and the test names given in all the appropriate instances are as she informed us,
  2. the test results of the spearman's correlation were also provided by our statistician.

Thank you once again!

Reviewer 4 Report

The main aim of this study was to evaluate the usefulness of FFFT in the functional assessment of physical fitness of HF patients, and to correlate FFFT results with CPET and echocardiography results as well as natriuretic peptide values. Regarding the authors, I would like to congratulate and thank them for their effort and motivation involved in this research study. The presentation of the research is well documented, with a scientific basis and respects the latest standards regarding the highest level scientific publications. The methodology was chosen correctly. The conclusions support and result from the research and open new directions for future research. The submitted work is interesting and exhausts the subject under discussion. I recommend the publication of this manuscript in its current version.

Author Response

Thank you for your kind review. We appreciate your opinion!

Round 2

Reviewer 1 Report

Thank you for the authors for their noticeable work and response to the reviewers comments. The authors have done a great job improving this manuscript. 

Just two comments/suggestion:

1. Would suggest to start the 'Discussion' section with the paragraph lines 165-169 instead of the current paragraph. 

2. Would also suggest adding to the conclusion what they found in relation to the echocardiogram findings and natriuretic peptide values that correlated with the FFFT assessments, as this will match their initial proposal/aim in the beginning.

Author Response

Thank you for your comments! We amended the text as suggested.

Reviewer 2 Report

Thank you for the revisions. Nice job.

Author Response

Thank you!

Reviewer 3 Report

I thank the authors for their efforts to improve the manuscript and for the implementation of all the corrections. 

Best regards

Author Response

Thank you!